# The Role of Intermittent Energy Restriction Diet on Metabolic Profile and Weight Loss among Obese Adults

**DOI:** 10.3390/nu14071509

**Published:** 2022-04-05

**Authors:** Agata Stanek, Klaudia Brożyna-Tkaczyk, Samaneh Zolghadri, Armand Cholewka, Wojciech Myśliński

**Affiliations:** 1Department and Clinic of Internal Medicine, Angiology and Physical Medicine, Faculty of Medical Sciences in Zabrze, Medical University of Silesia, Batorego 15 St., 41-902 Bytom, Poland; 2Chair and Department of Internal Medicine, Medical University of Lublin, Staszica 16 St., 20-081 Lublin, Poland; klaudiabrozyna19@gmail.com (K.B.-T.); wojciech.myslinski@umlub.pl (W.M.); 3Department of Biology, Jahrom Branch, Islamic Azad University, 7414785318 Jahrom, Iran; z.jahromi@ut.ac.ir; 4Faculty of Science and Technology, University of Silesia, Bankowa 12, St., 40-007 Katowice, Poland; armand.cholewka@gmail.com

**Keywords:** obesity, intermittent fasting, weight loss

## Abstract

Obesity is a disease defined by an elevated body mass index (BMI), which is the result of excessive or abnormal accumulation of fat. Dietary intervention is fundamental and essential as the first-line treatment for obese patients, and the main rule of every dietary modification is calorie restriction and consequent weight loss. Intermittent energy restriction (IER) is a special type of diet consisting of intermittent pauses in eating. There are many variations of IER diets such as alternate-day fasting (ADF) and time-restricted feeding (TRF). In the literature, the IER diet is known as an effective method for bodyweight reduction. Furthermore, IER diets have a beneficial effect on systolic or diastolic pressure, lipid profile, and glucose homeostasis. In addition, IER diets are presented as being as efficient as a continuous energy restriction diet (CER) in losing weight and improving metabolic parameters. Thus, the IER diet could present an alternative option for those who cannot accept a constant food regimen.

## 1. Introduction

Obesity is a disease defined by an elevated body mass index (BMI), which is the result of excess or abnormal accumulation of fat [1]. The data show that more than 600 million people worldwide were obese in 2015 [2]. The problem with excessive body weight is still growing, and in 2019, about 52.7% of the European Union’s population was overweight [3]. Obesity can lead to many chronic diseases such as type 2 diabetes, hypertension, and obstructive sleep apnea and can increase the risk of cardiovascular events. As well as this, it increases the risk of cancer for at least 13 parts of the body [4,5,6]. According to global data, in 2017, obesity was a cause of premature death for 4.7 million people [7]. Moreover, today, obesity has become the most important factor increasing mortality among patients with SARS CoV-2 infection [8]. 

Adipose tissue is known as an endocrine organ, consisting mainly of adipocytes, progenitor cells, and interstitial fibroblastic cells [9]. On the basis of different characteristics, adipose tissue is divided into subgroups. White adipose tissue (WAT) stores excess energy as fatty acids, while brown adipose tissue (BAT) is responsible for thermogenesis and has an influence on insulin sensitivity and glucose homeostasis [10,11]. “Beige” adipose tissue is another type, in which brown adipocytes arise within white adipose depots with thermogenic properties [12]. One of the most important factors derived from adipose tissue is adiponectin, which is responsible for the homeostasis of glucose and fatty acids. Adiponectin decreases glucose blood level by stimulation of glucose uptake by muscles, improvement of insulin sensitivity, or a reduction in hepatic glucose production [13]. In addition, regulation of lipid profile occurs through increasing the high-density lipoprotein (HDL) concentration and decreasing triglycerides [14]. Obesity is associated with an excessive amount of lipids, which are stored in adipocytes, resulting in their enlargement. Adipocyte capacity is limited, and exceeding the threshold results in molecular changes, dysfunction in adipocytes, and impaired production of factors derived from adipocytes [15]. Obesity triggers alterations in the quantity and quality of various types of cells that reside in adipose tissue, including adipose stem cells (ASCs). These alterations in the functionalities and properties of ASCs impair adipose tissue remodeling and adipose tissue function, which induces low-grade systemic inflammation, progressive insulin resistance, and other metabolic disorder [16].

Dietary intervention is fundamental and essential as the first-line treatment for obese patients, and the principal rule of every dietary modification is caloric restriction and consequent weight loss [17]. Yet, there are a wide variety of different diets, varying by the content of macronutrients; one universal diet does not exist. Intermittent energy restriction (IER) is a special type of diet consisting of intermittent pauses in eating (intermittent fasting (IF)). IER differs from continuous energy restriction (CER) by the specific time spent on fasting. There are many variations in IER diets such as alternate-day fasting (ADF) (Figure 1) and time-restricted feeding (TRF) (Figure 2); the main differences between them are the different times spent fasting and spent without restriction [18,19]. ADF consists of alternate fast days, three to four times a week, while TRF is characterized by food consumption in restricted hours with different variations in the duration of fasting time.

Fasting induces the expenditure of glucose circulating in the bloodstream and glycogen in the liver, along with a change in energy intake from glucose metabolism to fat-derived ketone bodies and free fatty-acid metabolism [20]. During fasting, fatty acids and glycerol are the main sources of energy. The liver is responsible for transforming fatty acids into ketone bodies, which are the main metabolic substrate of the brain. Ketone bodies play an important role in maintaining health and delaying aging processes, while they induce the production of many proteins, which regulate many signaling pathways such as polymerase 1 (PARP1), nicotinamide adenine dinucleotide (NAD+), peroxisome proliferator-activated receptor γ coactivator 1α (PGC-1α), and peroxisome proliferator-activated receptor α (PPAR-α). As a result, ketone bodies improve glucose regulation with abdominal fat loss and regulate cardiological parameters such as blood pressure and heart rate [20]. Moreover, fasting increases the antioxidant concentration and mitochondrial stress resistance. It is also responsible for inhibiting the mammalian target of the rapamycin (mTOR) protein-synthesis pathway. As a result, damaged cells, proteins, and mitochondria are more efficiently recycled [21].

If implemented among rodents with metabolic syndrome, it induces increased insulin sensitivity and a reduction in abdominal fat, blood pressure, and inflammation [22,23]. Moreover, in addition to normalization of the insulin level, among rodents with metabolic syndrome, the effect on leptin and adiponectin levels reverses their abnormalities [24]. In addition, IF has a beneficial effect on the gut microbiota, which can be essential to protect against the occurrence of metabolic syndrome [25]. Animal model studies with an IF diet group (especially ADF and TRF) and a control group fed ad libitum revealed lower body weights among the experimental group [26]. Furthermore, the IF diet had other health benefits among rats, such as prolonging their lives [27] and maintaining cognitive function during aging [28].

In humans, IF regimens have a beneficial effect on body weight and metabolic profile. In addition, fasting improves the availability of endogenous neurotransmitters and stimulates the exertion of endogenous cannabinoids and endorphins [23]. IER reduces proinflammatory and oxidative stress factors, and additionally, it improves insulin sensitivity and glucose uptake by cells [20]. The reduction in body fat was observed not only among obese participants but also among normal-weight men and women after three weeks of ADF [29].

## 2. The Influence of IER on Humans 

### 2.1. The Influence on Body Mass, Fat Mass, and Ectopic Fat

Body weight loss should be achieved mainly by reducing fat mass (FM) with the preservation of fat-free mass (FFM), which is metabolically active [30]. Yet, excessive reduction in FFM contributes to decreasing metabolism activity and, consequently, slowing body weight loss, potentially triggering a gain in weight [31]. 

In the literature, the IER diet is known as an effective method of body weight reduction (Table 1). The available trials are characterized by different durations, with the dietary intervention ranging from two weeks to as many as two years; however, most of the studies last 10–12 weeks. The reduction of circa 5 kg among IER diet groups is most commonly reported after 10–12 weeks of a trial [32,33,34], while shorter or longer durations of study have resulted in, respectively, smaller or greater weight losses [35,36]. There have been a few surveys in which an IER intervention did not induce a reduction in body mass weight; however, the possible explanation for obtaining such a result could be the short interval of the experiment or the small number of participants [37,38,39]. The participants in the mentioned studies presented BMI values that ranged from normal to overweight or obese. Surprisingly, the IER diets did not provoke compensatory food overintake on nondiet days; instead, they induced a consequent reduction of calorie intake by about 25% on nonfasting days [40,41].

The surveys based on implementing ADF among intervention groups consisted of periods called “fasting days” with severe energy restriction, with a usual intake of 25% of the energy required, and consequent alternating “feed days”, with no energy restriction or slight energy restriction and eating “ad libitum”. Varady et al. presented how an eight-week ADF diet induced a reduction of body weight by about 5.8 +/−1.1% among obese patients [42]. Moreover, subsequent research by the author reported that a 12-week ADF diet induced a decrease in FM while preserving FFM, which is essential for maintaining proper metabolism [43]. According to Soenen et al., the preservation of FFM depends not on the energy restriction but rather on the protein content of a diet [44]. Reduction of FM during the ADF diet was also reported by Harvie et al., but in this case, FFM was also reduced, even with an extra portion of protein-rich food [33]. Bhutani et al. conducted a trial comparing the effect of an ADF diet and exercise on body weight and different psychological aspects, in which patients were divided into four subgroups: diet group, exercise group, mixed group, and control [34]. Twelve weeks of intervention showed the greatest weight loss among the mixed group, with a similar reduction in body weight among the exercise and diet groups. Surprisingly, satisfaction and fullness were the highest among the diet group, while the feeling of hunger was the weakest among the diet group, with no change among the other groups. In every intervention group, the frequency of uncontrolled eating decreased.

Recent trials with TRF diets were characterized by a different length of the time-of-eating window, varying from 4 to 10 h per day in which participants could eat, and then they fasted for the rest of the day. Fasting periods usually start during the late evening or nighttime [45]. Implementation of TRF with 10 h of eating for 16 weeks induced 3.6% weight loss [46] and, for 12 weeks, induced 3.0% weight loss [47] among overweight, healthy individuals. In addition, according to self-assessment, a significant improvement in energy level and sleep satisfaction was observed [48]. Peeke et al. performed a trial with two subgroups on the TRE diet, differing from each other in the duration of the fasting period; the first combination consisted of 14 h of fasting, starting after dinner and lasting till 8 a.m., and the second had a 12:12 schedule [49]. Comparing two TRE subtypes followed for eight weeks did not demonstrate significant differences in body weight reduction between them, but in both groups, weight loss was significant compared to the baseline. In recent trials with 12 weeks on an 8 h TRF diet, the weight loss among obese women and men was about 2.6% at the end of the trial [41]. Shortening eating intervals to 4 or 6 h induced a similar reduction in body weight as in the studies mentioned above, i.e., 3% [50]. Moreover, according to Cienfuegos et al., there was no difference in body weight reduction between the groups with the 4 or 6 h TRF diet. 

Lately, it has been postulated that intermittent fasting (IF) can have an impact on ectopic fat, which might contribute to increased atherosclerosis and cardiometabolic risk. Ectopic fat depositions, defined as the accumulation of triglycerides within cells or sometimes around nonadipose tissues (i.e., liver, skeletal muscle, pancreas, heart), are at the center of metabolic health derangements [51,52].

In a study estimating the effects of alternate-day fasting vs. daily calorie restriction, a reduction in the visceral fat mass after 6 or 12 months of AMDF was shown compared to a non-food-restricted control group. However, this change was comparable to that of a continuous daily calorie restriction group [53].

Similarly, in another study [54], it was shown that 12 weeks of twice-weekly fasting or a low-carbohydrate high-fat diet were superior to the standard of care intervention in reducing hepatic steatosis in patients with nonalcoholic fatty liver disease, and no differences between the twice-weekly fasting and low-carbohydrate high-fat diets were observed.

### 2.2. Insulin Sensitivity and Glucose Tolerance 

Obesity and being overweight are connected with increasing insulin resistance. There are many indicators assessing insulin resistance that are widely available and easily calculate this from the concentrations of fasting glucose and insulin, such as HOMA-IR and Quicki [55,56]. Insulin resistance is the result of a chronic inflammatory state between adipose tissue and the liver, with elevated concentrations of IL-6 and TNF-alpha [57]. Furthermore, obesity induces a decrease in the synthesis and release of adiponectin, which is an insulin-sensitizing factor, while an IER such as ADF increases the concentration of adiponectin [58,59]. Furthermore, the ADF diet decreases the concentration of leptin, which is elevated among obese patients [60]. There are mixed results on the influence of IER on insulin sensitivity. Gabel et al. presented how a 12-week ADF diet lowered fasting hyperinsulinemia, suggesting a sensitizing effect of diet regimens [48]. A stable or decreased fasting insulin concentration without changes in the fasting glucose level was presented among patients with IER regimens in many scientific reports [33,61,62,63]. In the opposite case, 12 weeks of the TRE diet induced a reduction in the fasting glucose concentration without impacting the fasting insulin concentration [64]. Perhaps the greatest differences are in the results for the different characteristics of experimental groups. In addition, insulin resistance after one day of fasting depends on the sex; it is higher among women than men, probably caused by different physiological fasting adaptation processes [64]. It is believed that the influence of TRE on insulin and glucose levels may depend on the time of day with an eating gap [45,65]. Insulin resistance and the fasting glucose level were reduced among resistance-trained males without obesity when the eating gap was in the middle of the day [46,66]. On the contrary, fasting until the late afternoon or evening among resistance-trained men induced an increase in glucose concentration after dinner [67].

The IER diet significantly reduces the fasting glucose and HOMA-IR, as was shown in a meta-analysis including 12 intervention studies of at least one month in duration for a total of 545 participants [68].

### 2.3. Lipid Profile 

IER regimens induce an improvement in plasma lipids, such as a 15–35% reduction in triglyceride concentration and a 6–25% reduction in LDL concentration, with only a small effect on HDL concentration [41,69]. The IER diet has a beneficial effect on LDL by increasing its particle size, which is important for reducing cardiovascular risk, as a small particle size has more proatherogenic, prothrombotic, and proinflammatory properties [70,71]. Both 10 weeks of TRE among patients with metabolic syndrome [46] and 12 weeks of TRE among overweight patients [72] decreased the LDL concentration. However, there are contradicting data; in some cases, IER regimens did not influence the LDL level, as when Sutton et al. presented the results of a five-week TRE diet among obese or overweight men with prediabetes [73]. Perhaps the disparity is the result of different times of dietary intervention or the number of participants in each study. In most cases, IER slightly altered the HDL concentration; perhaps a longer time of dietary intervention is required, with additional endurance training, to significantly increase the HDL concentration [74]. 

### 2.4. Gut Microbiota

Some studies have found that after the Muslim holy month of Ramadan, where no food is consumed from dawn to sunset, subjects have increased levels of beneficial gut bacteria such as *Akkermansia*, *Butyricicoccus*, *Bacteroides*, *Faecalibacterium*, *Roseburia*, *Allobaculum*, *Eubacterium*, *Dialister*, and *Erysipelotrichi*. They also have increased microbial richness and diversity and increased levels of the beneficial short-chain fatty acid, butyric-acid-producing *Lachnospiraceae*. High concentrations of *Lachnospiraceae* are associated with a reduced risk of cancer, improved inflammatory bowel disease, better mental health, reduced allergies, and improved cardiorespiratory health [75,76,77].

In [78], in patients with metabolic syndrome, it was shown that the patients who had a stronger reduction in blood pressure were those who initially had lower levels of bacteria that produce the short-chain fatty acid; the change was proportionate, but these levels increased after fasting. However, by three months post-fasting, the propionate production had reverted almost to the baseline level, though the improvement in blood pressure remained, suggesting that the transient improvement in propionate production decreased hypertension via mechanisms beyond the gut (CD8+ effector T cells, Th17 cells, and regulatory T cells).

### 2.5. Biomarkers of Inflammation

In a large meta-analysis of randomized control trials (18 RCTs were included and a total of 700 participants) evaluating the effects of IF or ERDs on plasma concentrations of inflammatory biomarkers, it was demonstrated that IF regimens and ERDs may reduce CRP concentrations, particularly in overweight and obese subjects and with interventions lasting at least two months. However, neither dietary model affected the concentrations of tumor necrosis factor-α or interleukin-6 [68].

### 2.6. Hypertension

It is reported that IER regimens influence blood pressure among obese or overweight people. Many scientific reports prove that TRE has a particular ability to decrease systolic and/or diastolic blood pressure (BP) [50,73,79]. Furthermore, Kord-Varkaneh et al., in their systematic review and meta-analysis investigating the effects of IER on BP levels in a total population of 1400 participants, showed that both systolic and diastolic BP can be significantly reduced with IER [80]. Additionally, interventions lasting less than 12 weeks were found to be more effective than longer-lasting interventions, which could imply that compliance with IER might gradually fade and probably cannot be maintained for a long time, thus raising long-term efficacy concerns [81]. However, Chow et al. presented the opposite outcome, where TRE has no impact on blood pressure [64]. In the case of ADF, Gabel et al. and Stekovic et al. reported that the diet did not influence the systolic or diastolic BP [82]. The reduction in BP during IER regimens could be a result of improved vascular endothelial-dependent vasodilation [83]. 

### 2.7. Other Molecular Mechanisms

In [84], it was shown that CR reduces the percentage of ASC in the lin^−^ SVF while also reducing colony-forming ability. Therefore, it was postulated that CR appears to have antiproliferative effects on ASCs that may be advantageous from the perspective of cancer. CR also engages RNA processing of genes associated with a highly integrated reprogramming of hepatic metabolism [85]. CR reduces microsteatosis, decreases levels of superoxide anion, and increases protein expression of catalase and superoxide dismutase. Moreover, CR decreases lipofuscin staining, p21, p53, Acp53, and p16 but increases pRb/Rb and sirtuin-1 protein expression [86].

## 3. Adherence to IER Diets

Every effective dietary intervention has to be adherable if participants are to lose weight and improve metabolic parameters. Although long-term use of IER regimens has not been studied well, short-term IER regimens have been reported to have low dropout rates. TRF with 8–12 h of fasting is reported by many studies to be well-tolerated by patients, and surprisingly, the greater the adherence rate, the longer the period of fasting [47,50,64,73,79]. Studies reported 65–75% compliance during the trial period [33,50,62]. Most of the reports present a dietary intervention for 8–12 weeks; thus, further studies with a longer period of dietary regimens should be conducted. According to Gabel et al., the adherent rate during 12 weeks of TRF with an 8 h feeding period per day among healthy obese individuals was stable, and patients were able to follow TRF six days a week [79]. Furthermore, it was reported that TRF induced a significant reduction in daily eating duration from 11 to 8 h, which confirms the theory that IER regimens decrease appetite and calorie intake throughout the day/period without fasting [79]. As a result, the end of eating decreased from 7:30 to 6:00 p.m. after 12 weeks of intervention. Cienfuegos et al. reported that patients undergoing eight weeks of TRF, with a four- or six-hour feeding interval, also complied with the restriction for six days per week [50]. Yet, the adherence rate is difficult to evaluate; most of the studies base this on self-reports, which in some cases, could be inaccurate. Using objective methods of eating-time evaluation revealed mistakes in subjective self-reports [46]. Perhaps further studies with mobile app users should be conducted to avoid such mistakes. However, some findings using apps could also have limitations as participants are more likely to report healthy food than unhealthy [87].

## 4. Side Effects of IER Diets

The adverse effects of IER regimens seem to be rare and harmless, occurring mainly during the initial days of dietary intervention and affecting single people. Reported side effects include morning fatigue, headaches, suppressed and increased appetite with concomitant irritability, and dizziness [67,88]. IER diets are reported to have a mixed influence on mood and wellbeing. Reduction in mood disorders, anger, and tension [89] and mood improvement [90] has been reported among participants on IF regimens. On the other hand, Laessle et al. suggested that IER diets increase irritability and fatigue [39]. In addition, obesity commonly coexists with diabetes type 2. Reduction of caloric intake among patients with diabetes type 2 can improve glucose homeostasis and decrease body weight. However, appliance of the IER diet among patients on hypoglycemic drugs, such as insulin or sulfonylurea, may contribute to the increased risk of hypoglycemia. Events of hypoglycemia were presented among 35% of patients on insulin/sulfonylurea therapy [91]. Corley et al. presented that events of hypoglycemia occur mostly during fasting days or fasting periods, despite the proper reduction of drugs and education about hypoglycemia [92].

## 5. Comparison to CER Diet

Previous studies comparing the influence of IER and CER diets were performed mainly on healthy, but obese, individuals [93]. In most reports with short-term trials of IER diets or CER diets, both of them seem to have a similar impact on different metabolic parameters such as decreasing the levels of total cholesterol, LDL, triglycerides, fasting glucose, and insulin [33,94,95]. Furthermore, CER and IER appear to have had a similar impact on weight loss, reduction in BMI, and decrease in hip or waist circumferences [33,62,79,96,97]. In many surveys, weight loss between the IER and CER groups was similar, in the region of 4–8% [32,33,95]. The daily calorie restriction in the CER and IER diets remained similar in most of the studies mentioned. Calorie consumption on “nonfasting” days in many surveys ad libitum among IER diets induce equalization of restrictions to CER diets, where restrictions are present every day. However, there are some studies in which there was a disproportion between diets. Harvie et al. designed a study as a 25% energy restriction from estimated baseline energy requirements among both groups; however, the IER group during “fasting days” was asked to undertake a 75% energy restriction; thus, it seems that more a restricted diet was used in the IER group [33]. Ash et al. designed a study in which both IER and diet were isocaloric, averaging 1400–1700 kcal/day [97]. However, in the literature, there are a few reports that present different results, where CER induced greater body weight loss than IER [98]. Opposite results, where the IER diet resulted in a greater weight reduction among participants [99] were found, where the calorie intake among the IER group was approximately 130–200 kcal/day less than among the CER group [100]. A comparison of 8–12 weeks of an ADF diet and a constant calorie restriction diet (CER) for four weeks revealed a greater reduction in body mass among the CER diet group; however, FFM was preserved more among the ADF diet group [101]. On the contrary, Soenen et al. presented the view that the preservation of FFM between CER and IER is similar and depends not on the energy restriction but rather on the protein content of the diet [44]. Certain data suggest that, despite the lack of differences in summarizing weight loss among patients on CER and IER diets, the decrease in body fat mass is greater in IF regimens [33].

According to Sundfor et al., a long-term study comparing IER and CER regimens presented similar effectiveness of both diets in terms of weight loss, improving or maintaining cardiometabolic factors after one year of the trial, where both diets were based on similar caloric content [102]. Despite similar weight loss after six months for both groups, greater weight loss was observed in the IF group than in the CER group. Additionally, participants of the IER diet presented a stronger feeling of hunger. Keogh et al. conducted another long-term, one-year, randomized control trial to compare the CER diet with the IER diet, in which some patients were able to eat for one week ad libitum and others were restricted [96]. The CER group followed the 5500 kJ energy restriction continuously, while the IER group performed the same energy restriction for 1 week, followed by 1 week of usual diet; however, there were no significant differences between both diets [87]. In both intervention groups, the reduction of body weight was most significant during the first eight weeks of dietary intervention; however, weight loss was similar among groups, with no significant difference between them. Furthermore, the continuation of the dietary intervention until 52 weeks did not induce a further significant reduction in body weight in the CER and IER diets, and the level of reduction remained comparable in both groups. The main limitation of the mentioned study was the high dropout rate (40%) among participants during the initial eight weeks of the study, which could have impacted the results and their statistical analysis. Most studies conclude that the impacts of IER and CER on glucose homeostasis are comparable, while the levels of fasting insulin and fasting glucose are similar in both groups [33,62,99]. In view of similar caloric restriction in most studies between CER and IER diets, it seems that the positive impact of IER diets on body weight could be the result of a change in metabolism.

In the literature, there are only a few reports that compare the influences of both CER and IER regimens on mood and wellbeing, while the comparisons that are available present incompatible results. A greater number of participants who observed an improved mood was reported among the CER rather than IER group, with 46% vs. 32% [62]. On the contrary, the other study presented a lack of difference between the groups in the number of patients who observed a positive impact of the applied diet on their mood or behavior [33].

Meng et al., in their recent systematic review and meta-analysis estimating the effects of IER and CER on the lipid profile, postulated that both dietary approaches reduce the levels of total cholesterol, LDL cholesterol, and triglycerides but have no effect on HDL cholesterol. CER was associated with greater reductions in the total cholesterol level compared to IER, and greater reductions in lipid markers were related to higher baseline levels. Furthermore, when the caloric reduction was greater than 50%, no significant reductions in total and LDL cholesterol levels were observed [103].

## 6. Current Limitations of Knowledge on IER Diet and Future Directions

Although there is some evidence to suggest that IER has beneficial effects on human health, there are still some issues that warrant further investigations. First, high-quality, strong scientific evidence regarding the long-term effects of IER is limited since the vast majority of studies had a duration of fewer than 6 months. Second, it is not yet known whether IER protocols are a safe recommendation for the general population and specific populations, including males, the elderly, and patients with morbid obesity and diabetes mellitus, as well as children or adolescents. Third, what is the impact on people’s health of long-term use of this diet? Furthermore, the most important concern is to determine who will benefit the most from an IER intervention, depending on their personality traits and, most importantly, comorbid health conditions [81,104].

## 7. Conclusions

The dietary intervention induced by IER involves a significant reduction in body weight and a decrease in waist and hip circumferences (Figure 3). Furthermore, improvements to the control of cardiovascular risk factors such as lipid profile and blood pressure are also presented. Beyond this, fasting has beneficial effects on the adipose tissue, while IF restores the balance between leptin and adiponectin production among obese patients. The influence of IER on glucose homeostasis is inconsistent; however, it appears to have a potential influence on glucose and insulin levels. In recent trials, most reports presented similar effectiveness levels of the IER and CER diets for losing weight and restoring the metabolic balance; yet, there are no recommendations that suggest choosing any of the diets over others. IER could represent an effective alternative to dietary interventions for those who are not able to live with constant dietary austerity. Further, robust studies are required to confirm the safety and long-term effect of using this diet.

## Figures and Tables

**Figure 1 nutrients-14-01509-f001:**
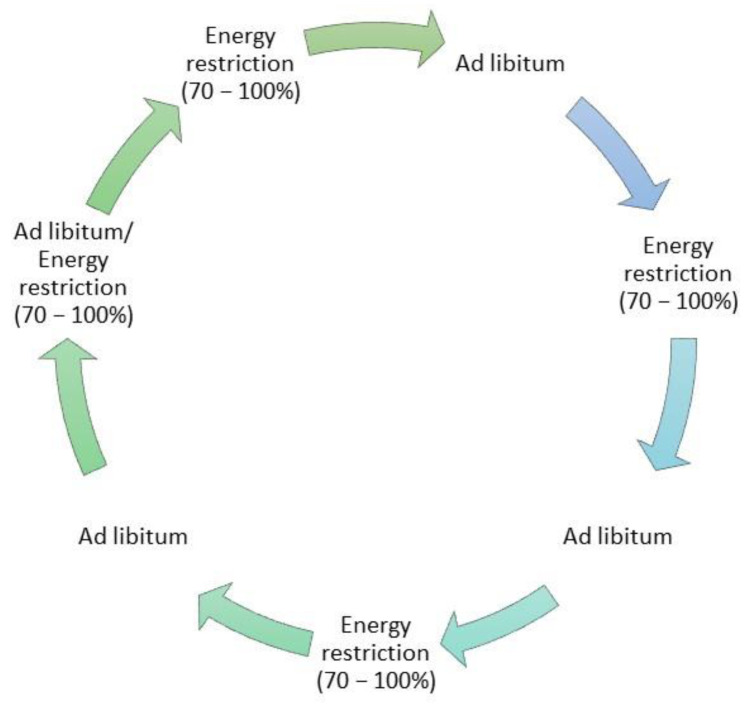
Exemplary scheme of alternate-day fasting (ADF) diet.

**Figure 2 nutrients-14-01509-f002:**
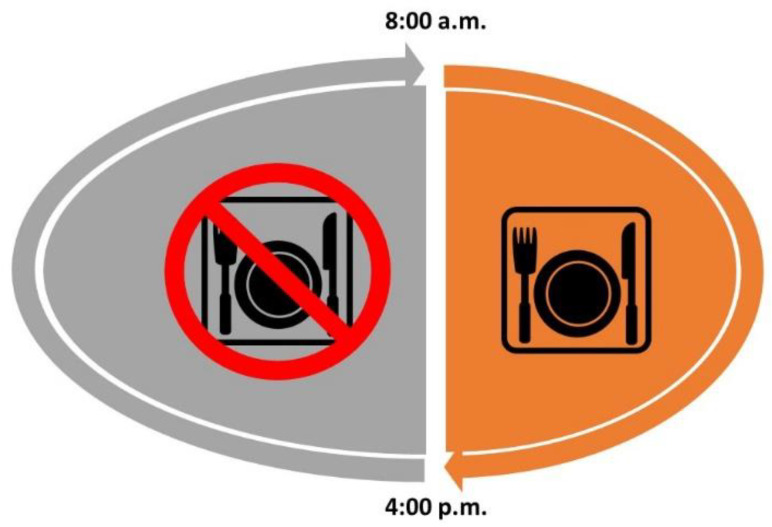
Graphical presentation of time-restricted feeding (TRF) diet.

**Figure 3 nutrients-14-01509-f003:**
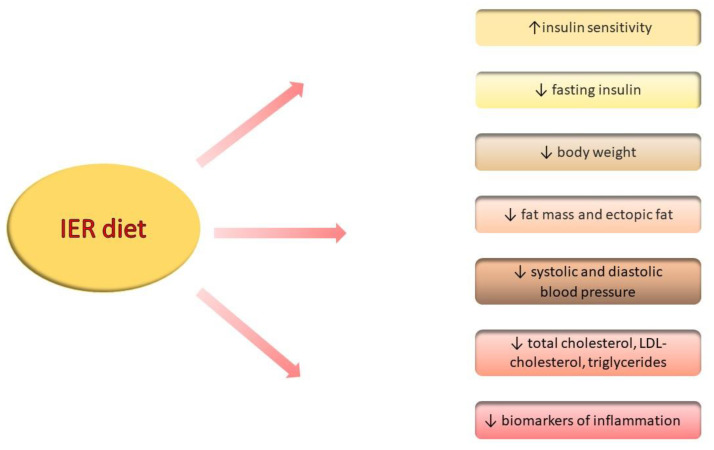
The effect of IER diet on different metabolic and inflammatory parameters (↑—increase, ↓—decrease/reduction).

**Table 1 nutrients-14-01509-t001:** Characteristics of included studies on body weight changes among patients on diets based on intermittent energy restriction (IER) such as alternate-day fasting (ADF) and time-restricted feeding (TRF).

ADF					
	Characteristic of Group	Dropout Rate	Composition of Diet	Time of Therapy	Effect on Body Weight
Heilbronn et al., 2005 [25]	16 patients with BMI ranging from 20 to 30 kg/m^2^	-	Fasting days had 0% of energy intake, doubling the energy need on nonfasting days	3 weeks	Reduction of FM and FFM
Halberg et al., 2005 [27]	8 overweight males	-	Fasting days had 0% of energy intake for 20 h, with ad libitum intake on feeding days and at all other times	2 weeks	Lack of bodyweight reduction
Varady et al., 2009 [31]	16 obese patients: 12 females; 4 males	-	Fasting days met 25% of energy needs, and the following days were ad libitum	8 weeks	Reduction of BW of 5.8 kg +/−1.1 kg
Varady et al., 2013 [32]	12 overweight/obese males and females; 15 controls	7% IER 7% control	Fast days had 25% of energy intake, and the following days were ad libitum	12 weeks	Reduction of body weight and FM, FFM with no change
Harvie et al., 2013 [22]	75 overweight/obese females (IER-A: 37; IER-B: 38)	11% IER-A 26% IER-B	IER-A: fasting days had 30% intake of energy needs for 2 days/week and CER diet for 5 days/week IER-B: fasting days had 30% of energy intake plus 250 g of protein-rich food and CER diet for 5 days/week	17 weeks	Similar reductions of body weight, FM, and FFM in both groups
Bhutani et al., 2013 [23]	Obese male and females: 25 IER; 18 IER + EX; 24 EX; 16 controls	36% IER 33% EX 11% IER + EX	IER fasting days met 25% of energy needs, and the following days were ad libitum EX: 3 times/week	12 weeks	Reduction of body weight in every intervention group: IER + EX (6 ± 4 kg) > IER (3 ± 1 kg) = EX (1 ± 0 kg)
**TRF**					
	**Characteristic of Group**	**Dropout Rate**	**Composition of Diet**	**Time of Therapy**	**Effect**
Gill et al., 2015 [35]	8 participants overweight/obese: 5 males; 3 females	-	10 h eating period including nonwater beverages, with 14 h fasting window per day	16 weeks	Reduction of body weight by about 3.6%
Wilkinson et al., 2020 [36]	19 participants with obesity: 6 females; 13 males	-	10 h eating period, with 14 h fasting window per day	12 weeks	Reduction of body weight by about 3%
Peeke et al., 2021 [38]	79 participants: 39 on TRF 12:12; 29 on TRF 14:10	30% Group 1 30% Group 2	Group 1: 12 h eating period, with 12 h fasting period per day Group 2: 10 h eating period, with 14 h fasting period per day	8 weeks	Reduction of body weight by about 7.1% among Group 1 and about 8.5% among Group 2; the difference was not statistically significant
Gabel et al. [37]	23 participants with obesity	-	8 h eating window, with 16 h fasting period per day	12 weeks	Reduction of body weight by about 2.6 +/−0.5%
Cienfuegos et al. [39]	58 obese participants: 19 in Experimental Group 1; 20 in Experimental Group 2; 19 in control group	Experimental Group 1: 5% Experimental Group 2: 15% control group: 26%	Experimental Group 1: 4 h eating window, with 20 h fasting period per day Experimental Group 2: 6 h eating window, with 18 h fasting window per day	10 weeks (2 weeks of body weight stabilization and 8 weeks of TRF)	Significant reduction of body weight among both intervention groups compared to controls: 3.2 +/−0.4% weight loss among Groups 1 and 2

Note: BMI, body mass index; FM, fat mass; FFM, free fat mass; CER, continuous energy restriction; EX, exercise.

## Data Availability

We used PubMed and web of science to screen articles for this narrative review. We did not report any data.

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
