# Peer review of "The Role of Intermittent Energy Restriction Diet on Metabolic Profile and Weight Loss among Obese Adults"

_nutrients, 2022, doi:10.3390/nu14071509_

Round 1

Reviewer 1 Report

This is a strong and useful paper.  My only suggestion on your section comparing calorie restriction to IF is to see if you can discuss the calorie differences between the trials.  In other words, in studies where there are no differences between the two, did both end out calorie restricting to a similar amount?  And in studies where one was better than the other was it driven by calorie restriction?  Id also like you to discuss if intermittent fastings effects are tied to the fast itself independent of calorie restriction or is it primarily driven by a decrease in calories?  If you can address these in the paper I feel it can be accepted 

Author Response

Please find  the reply as attachment

Reviewer 2 Report

Comments

The authors documented for “The role of intermittent energy restriction diet on metabolic profile and weight loss among obese adults” Many diseases in adults are directly or indirectly associated with obesity. Because of this reason, many people have been interested for decreasing methods of their weight. Among the many methods, the intermittent energy restriction diet is known as the most effective method. For improvement of this method, metabolic profiles from this method are very important information. This review study will be is available information to improve as the method without side effects. Entirely, this manuscript is well written, however, to be more excellent manuscript, authors should consider to apply my comments.

  1. Entirely, this manuscript has insufficient molecular information in several cells including adipocyte, adipose derived stem cell and hepatic cells for clinical application.
  2. Based on some studies, IER increase a risk for hypoglycemia, diabetes mellitus (type 2). The authors should add information for the molecular and nutrient in these diseases in the side effects section.  

Author Response

Please find  the reply as attachment

Round 2

Reviewer 2 Report

no more comments